# Mental Gravity: Depression as Spacetime Curvature of the Self, Mind, and Brain

**DOI:** 10.3390/e25091275

**Published:** 2023-08-30

**Authors:** Lachlan Kent

**Affiliations:** Mental Wellbeing Initiatives, Royal Melbourne Institute of Technology, Melbourne, VIC 3001, Australia; lachlan.kent@rmit.edu.au

**Keywords:** spacetime, self, depression, cognition, gravity, default mode network

## Abstract

The principle of mental gravity contends that the mind uses physical gravity as a mental model or simulacrum to express the relation between the inner self and the outer world in terms of “UP”-ness and “DOWN”-ness. The simulation of increased gravity characterises a continuum of mental gravity which states includes depression as the paradigmatic example of being down, low, heavy, and slow. The physics of gravity can also be used to model spacetime curvature in depression, particularly gravitational time dilation as a property of MG analogous to subjective time dilation (i.e., the slowing of temporal flow in conscious experience). The principle has profound implications for the Temporo-spatial Theory of Consciousness (TTC) with regard to temporo-spatial alignment that establishes a “world-brain relation” that is centred on embodiment and the socialisation of conscious states. The principle of mental gravity provides the TTC with a way to incorporate the structure of the world into the structure of the brain, conscious experience, and thought. In concert with other theories of cognitive and neurobiological spacetime, the TTC can also work towards the “common currency” approach that also potentially connects the TTC to predictive processing frameworks such as free energy, neuronal gauge theories, and active inference accounts of depression. It gives the up/down dimension of space, as defined by the gravitational field, a unique status that is connected to both our embodied interaction with the physical world, and also the inverse, reflective, emotional but still embodied experience of ourselves.

## 1. Mental Gravity: Depression as Spacetime Curvature of the Conscious Self, Mind, and Brain

The Temporo-spatial Theory of Consciousness (TTC) maintains that there is a correspondence between the spatial and temporal organisation of the brain and the spatial and temporal structure of experience [1]. The principle of mental gravity [2] maintains that a key dimension of space also gives structure and meaning to the conscious experience, namely the up/down dimension associated with the gravitational field. This correspondence between the two theories goes beyond the metaphorical notion of “CONSCIOUSNESS IS UP, DEATH IS DOWN” [3]. This paper explores a deeper between the two theories in terms of the structure of subjective experience (phenomenology), the structure of thought (cognition), and the structure of the brain (neurobiology). This exploration is therefore aligned with Northoff’s related ideas of: (1) the “common currency” of space and time (i.e., spacetime, spatiotemporal, or temporospatial) across all levels of psychological explanation (i.e., phenomenological, cognitive/behavioural, and neurobiological); and (2) the “spatiotemporal psychopathology” of depression and other mental disorders, where the disordered experience of space and time is associated with disordered spatial/temporal binding and the integration of neural activity.

The specific idea to explore is the possibility that spatial dimensions and orientations are not equivalent in experience and therefore, as adduced via the “common currency” approach, neither should we expect them to be equivalent in other aspects of cognition or even neurobiology. The proposition is that the vertical dimension has a special status owing to its relation to physical gravity and that being “down” psychologically connotes meanings specific to that status. The principle of mental gravity posits that psychological up-ness and down-ness represent a primary, universal, and core dimension of consciousness that provides the TTC a more central role in the multidimensional characterisation of consciousness in terms of local and global states, content-related and functional dimensions, and so on [4]. While the “levels” of consciousness may not be a useful conceptualisation, the idea of “more” or “less” conscious states in a unidimensional or multidimensional space remains a plausible way to categorise various disorders of consciousness in contrast to the “full” consciousness of alert wakefulness [5]. Depression is put forward as a canonical experience of psychological down-ness, which may also be considered a disorder of “diminished” consciousness in a multidimensional framework [6].

The current paper argues that the distorted experiences of the self in time and space in depression can be modelled via the TTC as a marker of disordered conscious experience. The key focus will be the experience of subjective time dilation, which is the experience of a reduced temporal flow of consciousness [7,8], an experience that is highly prevalent in those diagnosed with depression [9]. The principle of MG connects the experience of subjective time dilation to the dilation of time in physics under conditions of gravity. The core feature of the MG model is the distortion or *curvature* of subjective time and space, which manifests in many ways in depression including at the neurobiological level in terms of the spreading of neural activity through the brain’s spatiotemporal organisation [10]. 

The paper is structured as follows: (1) the principle of MG is summarised; (2) the physics of gravitation are systematically related to the mental *simulation* of gravity-like behaviours; (3) MG spacetime curvature is illustrated in depression; (4) a model for the curvature of the brain’s spacetime is presented; (5) similarities between MG and the predictive processing accounts of curvature are described; and (6) implications for the TTC are summarised.

## 2. Overview of the Principle of Mental Gravity

Gravity is a fundamental law of nature in physics [11,12] but, beyond physics itself, “gravity models” have been used in economics and other special sciences to describe the movement of goods, services, people, or other economic or biological factors [13,14]. The current paper seeks to explore a gravity model for psychology and consciousness science—to make an explicit connection between the mathematical models of physical gravity and the internal gravity model of cognitive neuroscience [15,16,17,18].

Internal models are a neuroscientific explanation for how humans (and primates) move in their environment according to, among other things, physical laws of motion [19]. This includes modelling gravity according to implicit (Newtonian) physics [20], a model that can differentiate linear acceleration from the downward pull of gravity [21]. The aim of this paper is to outline a principle of mental gravity (MG) that explores the relation between the internal gravity model to broader aspects of cognition, consciousness, selfhood, and psychopathology. 

Before delving further, it is essential to clarify what the principle of MG unequivocally does not entail. First and foremost, MG should not be misconstrued as a sociological “law of attraction” that posits some spiritual or mental force binding individuals together. Additionally, MG is distinct from the quantum gravity effects proposed in the theory of Orchestrated Objective Reduction [22] involving microtubules. Furthermore, it does not align with the religious concept of a “fallen” human nature in any moral or mythical sense. Instead, the principle of MG solely establishes a connection between physical gravity and fundamental mental phenomena associated with feelings of being down, low, heavy, and slow, within the domains of affect, personality, and psychopathology. By clarifying these distinctions, we ensure a precise understanding of the scope and focus of MG, rooted in empirical observations and theoretical considerations of gravitational experiences and their psychological implications.

The principle of MG applies to phenomena associated with being down, low, heavy, and slow, particularly those evident in the experience of depression. These phenomena can manifest as concrete, embodied, physiological experiences such as the (mis)perception of increased bodily heaviness [23,24,25] or psychomotor retardation (i.e., slowed speech, decreased movement, and impaired cognitive function) [26,27,28]. Or they can manifest in very abstract, metaphorical representations of depression as a personal descent to a low or “fallen” state of being, which includes feeling stuck or weighed down by guilt, worthlessness, isolation, despair, and dislocation from the world [29,30].

### 2.1. Continuum of MG Experiences

Depression is not the only mental state associated with being down or low, but it is a more global description of mood that subsumes other negative emotional or feeling states like sadness [23], grief [31], boredom [32], frustration [33], and even nihilism [25]. Nor is depression the only way that MG can manifest in an experience or behaviour. As explained below, there exists a hypothetical continuum of MG states ranging from positive “up” and “down” states like joy and calmness, respectively, to negative “up” and “down” states like anxiety and depression, respectively. The principle of MG focuses on depression but does not rule out a whole spectrum of gravity-like experiences depending on the precise context and content of any given mental state. There are an indeterminate number of intermediate states along this continuum, of course, in the same way that there are infinite combinations of vertical and horizontal movements or asymmetries within the gravitational field [34].

The principle of MG reflects our experience of physical gravity in that it is not just about vertical elevation (UP/DOWN) but also horizontal alignment (ALIGNED/UNALIGNED) to the gravitational field. The abstract physical concept of falling, for example, involves a simple change in vertical position from high (UP) to low (DOWN). With respect to our *contextualised* experience of falling as embodied objects, however, we tend to fall when our centre-of-gravity is too high for the degree of non-alignment between our body’s axis and the gravitational axis (i.e., when we are not perpendicular to the Earth). A fall also usually involves a bodily change from a vertical to horizontal alignment, from standing to some degree of lying down, in other words. The principle of MG therefore models the full range of MG experiences by combining the feelings of both vertical elevation and horizontal alignment.

### 2.2. Anxiety and Depression as Distortions of Lived Space and Time

Although much of this paper focuses on verticality, the inclusion of horizontal alignment means the principle of MG relates to the whole four-dimensional field of lived space and time. This is in keeping with the longstanding analyses of lived space and time in mental disorders like depression and anxiety [35]. In the MG formulation, anxiety is an elevated but unaligned state analogous to the fear experienced when one is at-risk of falling, about to fall, or in the process of falling. Depression, on the other hand, is an unelevated and unaligned state analogous to the pain or discomfort (including negative self-worth or shame) experienced when one has fallen. There are many phenomenological corollaries that accompany these falling versus fallen states, especially in depression which is often described in experiential or metaphorical terms as “DEPRESSION IS WEIGHT”, “DEPRESSION IS CAPTOR”, “DEPRESSION IS ENCLOSED SPACE”, “DEPRESSION IS FORCE”, and “DEPRESSION IS DARKNESS” [36]. Being stuck or weighed down at some lower *depth* by various feelings of guilt, worthlessness, and despair also comes with feelings of isolation, detachment or dislocation from the world [30]. All of space is distorted, in other words, not just the vertical dimension. This is because gravity also defines centrality in addition to verticality. Not that being depressed is straightforwardly akin to being self-centred, but there is a tendency to attend more to self-related information and less to externally related events [37].

### 2.3. The Internal Gravity Model

The basis for the principle of MG is the observation that gravity structures how we think about ourselves, others, and the world. Humans have very firm perceptual and motor expectations about how objects will behave under the conditions of gravity, including our own body as the primary object of our experience [17]. This internal gravity model is mediated by the vestibular system, which integrates multimodal gravitational input from vision, proprioception, and viscera [15,38]. In addition to being the “balance system”, the vestibular system also mediates behavioural functions like posture, gait, and gaze, as well as autonomic functions and higher-order cognitive processes like spatial navigation, learning and memory, and behavioural control [18,39,40].

The internal gravity model’s strong expectations can alter behavioural strategies based on the body’s alignment to the gravitational field. Standing upright motivates novel, exploratory behaviours over repetitive, exploitative behaviours [18]. We also have aesthetic preferences for vertically aligned objects in art [41] and, more importantly for the present analysis, the internal gravity model also affects the phenomenological experiences of the self as anchored to the body [42], egocentrism [43], perceived body weight [44], and the perception of time and space [45,46].

The internal gravity model models complex interactions between the body and the external environment by integrating vestibular signals with other sensory inputs from vision, proprioception, and viscera [15]. Multisensory integration is performed via the insular cortex, with the posterior insular cortex operating when there is graviceptive feedback from the vestibular system, and the anterior insular cortex operating when there is no graviceptive feedback—in other words, the anterior insular cortex mediates imagined or simulated graviception [16]. This distinction between posterior and anterior insula activity when simulating or imagining the gravity’s effects is key to understanding how physical gravity “becomes” mental—i.e., it is one of the neural correlates of MG. 

### 2.4. Mental Gravity as Simulated Graviception

The anterior insular cortex mediates MG, not in response to physical cues from the external environment, but from psychological cues derived from the internal milieu. Under the conditions of imaginary or simulated gravity, the graviceptive system is freed from the constraints of physical and physiological input and can instead respond to non-physical, social, or emotional stimuli, including the self. The anterior insular cortex is a key salience network hub associated with interoceptive subjective feelings [47], social emotions [48], emotional awareness [49], and empathic pain perception [50]. As such, MG simulation is used to communicate subjective feelings about a person or situation. In other words, MG simulates the effects of physical gravity to communicate the internal, autonomic, and interoceptive predictions or inferences regarding affective, allostatic, and emotional states [51,52].

### 2.5. Universal Verticality Value

The MG simulation responds to social and emotional value. As with languages and metaphors across different cultures [53,54,55,56], verticality defines a universal aspect of intrinsic human value—hereafter called Universal Verticality Value (UVV). Adapting to gravity has evolutionary survival value, where the tendency to fall and feel heavy is factored into everything we do. Graviception (i.e., the perception of gravity) also structures key aspects of language, thought, and action. Being “UP” denotes positive emotional valence across cultures and contexts [53]. Some African languages use “UP” metaphors to communicate joy, certainty, courage, or social value (i.e., pride), and “DOWN” metaphors for lower social value (i.e., shame), calmness, or comfort [55,56]. The first “orientational” English metaphors proposed by Lakoff and Johnson [54] are “HAPPINESS IS UP, SAD IS DOWN”, “CONSCIOUSNESS IS UP, UNCONSCIOUSNESS IS DOWN”, “HEALTHY AND LIFE ARE UP, SICKNESS AND DEATH ARE DOWN”.

### 2.6. UVV and the Continuum of MG experiences

The continuum of MG experiences therefore reflects a higher or lower UVV which, as stated above, is not just about being “UP” or “DOWN” in the gravitational field. It also depends on whether that “UP” or “DOWN” state is vertically aligned or unaligned. Alignment to the gravitational field indicates positive value [41] for MG because it denotes stability and thus a higher UVV, regardless of whether the height is “UP” or “DOWN”. If the position in the gravitational field is “UP” and vertically aligned, then the experience is positive (e.g., joy, excitement, and ecstasy). If the position in the gravitational field is “DOWN” and vertically aligned, then the experience is also positive (e.g., contentedness, calmness, and relaxation). But if the position in the gravitational field is “UP” and not vertically aligned, then the experience is negative and akin to falling (e.g., fear, insecurity, and panic). If the position in the gravitational field is “DOWN” and also not vertically aligned, then the experience is also negative and akin to having fallen (e.g., hopelessness, dejection, and sadness). The difference between falling and having fallen is similar to the distinct but related states of anxiety and depression, respectively, and so MG can model anxiety and mood disorders in terms of a common explanatory mechanism. However, depression is the iconic experience of MG because it is associated with being down, low, heavy, and slow, meaning it is associated with “gravity-like” behaviours.

### 2.7. Feeling and Acting “As Though” Affected by Strong Gravity

Imagine gravity was much stronger. Imagine you felt heavier and moving was more difficult, even breathing or talking. Imagine standing was an enormous effort that left you mentally or physically exhausted. In strong gravity, you would engage in fewer novel behaviours to explore your environment. You would instead need to conserve energy through slow, minimal, and deliberate movements; a downcast posture (lowered head, eyes, and shoulders); a softer and deeper tone of voice; and other behaviours that reflect a weighty environment. The principle of MG posits that these same “gravity-like” behaviours manifest when people are depressed or, to a lesser extent, anxious. Symptoms of anxiety and depression are both associated with less-directed exploratory behaviour, where individuals are less motivated to seek further information and so exhibit maladaptive avoidance behaviours [57]. A non-alignment to gravity (i.e., lying supine) is associated with a tendency to exhibit more “exploit” and less “explore” behaviours [18]. This implies that MG makes people *act as though* something is preventing them from exploring their environment, namely physical gravity. This principle can explain basic observations such as why depressed individuals tend to have a slumped posture [58], and anxious individuals tend to have problems with balance [59]. It is as though they are experiencing stronger physical gravity which, especially in the case of depression, leads to behaviours that are stereotypically down, low, heavy, and slow. People such as astronauts in hypogravitational or weightless conditions also move slowly, but this is due to motor control factors that maintain smoothness and accuracy [60], and is therefore not a counterexample of the motor slowing under the conditions of MG.

### 2.8. Summary of Mental Gravity

MG involves *simulating* stronger or misaligned physical gravity in order to communicate UVV in a meaningful, heuristic way through gravity-like behaviours. The intuitive physics of gravity serves as a mental model or simulacrum for MG, such that individuals communicate their emotional or dispositional response to the *social* or *emotional* world by simulating the gravitational relationship between their body and the *physical* world. The principle of MG draws a behavioural analogy between the physical and social–emotional environments.

## 3. Neural Correlates of Mental Gravity

Via the anterior insular cortex, the salience network mediates the expression of MG by virtue of other roles mediating interoceptive inference, emotional awareness, and the simulation of the internal gravity model [16,48,61]. The salience network’s more general function is to “gatekeep” the various aspects of executive control [62] by switching the attentional resources between self-related information processing of the Default Mode Network (DMN) and task-related information of the Central Executive Network [63]. The DMN’s general role is to create the self’s internal narrative through episodic and autobiographical memory, social cognition, language, and semantic memory [64]. Davey and Harrison [65] also dubbed the DMN as the “brain’s centre of gravity” given its importance for self-referential processing, which is similar to the philosophical ideas presented by Dennett [66] of the self being the “centre of narrative gravity”. So, whereas the salience network mediates the expression of MG, the DMN mediates the role of the self in forming evaluations of UVV. The DMN creates the conditions for MG and the salience network creates the mechanism of simulating and thus embodying the effects of physical gravity.

### Mental “Mass”

Whereas physical gravity requires physical mass (e.g., the Earth), MG revolves around the accumulated “mass” of personal experiences and autobiographical memory as represented by the DMN self-narrative [64]. The *mental mass* of the DMN’s internal narrative places the self at the phenomenological centre of simulated MG in the same way that non-simulated graviception anchors the self to the body [42] to give an egocentric perspective [43] to perceive time and space [45,46]. The subjective experience of space and time are therefore intimately connected to the DMN’s composition of mental mass, with more mass being equivalent to stronger gravity that then translates into the mental states associated with being down, low, heavy, and slow. In depression, self-related autobiographical memory tends to be non-specific [67,68], over-general [69,70], intrusive [71], and emotionally negative [72,73,74]. This distortion of autobiographical memory and the narrative self is the MG’s source of the disturbed experience of space and time that speaks directly to the TTC [1]. 

## 4. Translating Basic Physics into a Psychological Gravity Model

Two physical theories are used to model or explain gravity (or more formally, “gravitation”). Newtonian [12] gravity involves attractive forces between two masses and is used to model our direct experience of the Earth’s gravity as well as astronomical phenomena such as microgravity in space, planetary orbits, and so on. Einstein’s [75,76,77] general theory of relativity is a more precise mathematical model of the curvature of spacetime, which can also explain more exotic gravitational phenomena like black holes, time dilation, and the bending of light. However, despite its precision and objective truth, general relativity is not part of our intuitive physics of how biological objects move in the Earth’s relatively weak gravitational field. Relativistic effects are only evident in very strong gravitational fields, in high-precision instruments, or at velocities near to the speed of light.

The principle of MG can also be formulated in two ways: (1) according to Newton’s [12] theory to model the “intuitive physics” [78] of feeling mentally and physiologically “down”; and (2) general relativity as an isomorphic process of mental phenomena that affects deeper, more fundamental mental processes that are only evident in highly precise (i.e., clinical research) contexts where MG is very strong or prolonged. The current exposition of MG will therefore address both the Newtonian and Einsteinian formulations of gravity in turn, with intuitive physics explaining more basic properties of being down, low, heavy, and slow, and fundamental physics explaining the more implicit or extreme aspects of MG in depression. Table 1 attributes the different types of MG experience to the different levels of explanation. The postulate of the principle for intuitive physics is that the physical mass of the body is perceived to increase in MG simulation due to the increased mental mass.

DMN activity occurs at the brain’s narrative centre of gravity. To translate this into Newtonian physics, a gravitational field is defined via two properties: (1) how large the interacting masses are; and (2) how close they are to one another. The strength of the gravitational force between two masses is determined via Newton’s [12] inverse-square law of gravity, as shown below:(1)$$F=G\frac{m_{1}m_{2}}{r^{2}}$$
where the force of gravity (*F*) increases proportional to the mass (*m*_1_ and *m*_2_) and decreases proportional to the distance (*r*) as a function of the gravitational constant (*G*). Distance, the Earth’s gravity, and the gravitational constant are not relevant here because our intuitive physics does not include the direct experience of changes in the Earth’s mass, significant distance from the Earth’s centre of gravity (unless one counts recent astronautical experience), or changes in fundamental physical constants. All we experience is a change in physiological mass and so this can be the only intuitive basis for MG simulation that makes depressed individuals feel heavy when they also feel down, low, and slow. 

A heavier body requires extra effort (i.e., energy) to move the body a given distance through the environment, or else it will take longer to move the heavier body with the same amount of energy. These cognitive calculations can be modelled using simple physics. The equation for overcoming gravitational force (Equation (1)) is:(2)E=mgH
where potential energy (*E*) is created when you raise a mass (*m*) to a height (*H*) in the Earth’s gravitational field (*g*), which is in turn calculated by:(3)$$g=G\frac{M}{r^{2}}$$
where *M* equals the Earth’s mass in kilograms, *r* the Earth’s radius in metres, and *G* the gravitational constant (i.e., the strength of the gravitational force acting on a body of a given size and mass over a given amount of time, 6.67 × 10^−11^ m^3^ kg^−1^ s^−2^). If the increased mental mass in depression increases the expected weight of the body’s physical mass, then the effect on Equation (2) is that the same amount of energy (*E*) will raise the body to a lower height (*H*) or, equivalently, more energy will be required to raise the larger mass to the same height. In effect, it is energetically more “costly” for someone to move while depressed. 

These equations show how depression can cause someone to feel heavy on account of being down or low, but it does not show how they could also feel slow. The physics of “power” (*P*) calculates the amount of energy exerted in a given amount of time (*t*): (4)$$P=\frac{E}{t}=\frac{mgH}{t}$$
where the time taken to raise a mass (*m*) to a given height (*H*) depends on the amount of energy (*E*). If the mass of the body increases, then the same amount of energy will take longer to counteract the force of gravity, making movement slower and/or more effortful—both of which are symptoms of depression. Psychomotor retardation is a common symptom of depression with severe impacts on daily functioning due to slowed speech, decreased movement, and impaired cognitive function [27]. From feeling down and low due to the increased mental mass, simple physics can also explain why depression makes people feel heavy and move slowly.

## 5. The Einsteinian Mechanics of Non-Intuitive MG in Depression

However, this mechanical power calculation does not explain every aspect of “feeling slow” in depression. In addition to moving slowly on account of the increased energy costs, weight, or power calculations, depression also tends to lead to a more abstract feeling that *time itself* has slowed down, which is an experience called “depressive time dilation” [79,80]. This equates to a reduction in subjective temporal flow or the reduced rate of felt temporal passage [8] which, in depression at least, is distinct from the estimations of duration like those in the power calculation above. A meta-analysis of clinical patients in six separate studies has confirmed (Hedge’s *g* = 0.66, medium effect size) that severely depressed individuals tend to experience subjective time as slow or dilated [9]. This is despite the fact that, according to the same meta-analysis, no deficits in interval timing were clearly evident from the 16 separate studies of over 400 patients. “Slow” time does not equate to “more” time, in other words. The story is more complicated because subjective time is a complex scientific question. Even though the body moves more slowly, and conscious experience is felt to unfold more slowly, clock time is mostly judged accurately while depressed, and so the MG calculations above in relation to energy, weight, and power do not generalise to all the aspects of temporal cognition. Psychomotor retardation in depression is not equivalent to depressive time dilation, and so there must be a different explanatory model.

Phenomenological psychopathology asks more open, qualitative questions about temporal experiences than the typical psychophysical time perception tasks like interval estimation, production, reproduction, discrimination, and bisection, or temporal processing tasks like temporal order, simultaneity, or gap detection [8]. Depressed individuals describe their experience of time using phrases like “Time seemed an eternity”, “Time seems to drag”, “Time is void”, “Time is stopped”, and “I lost flow of time” [80]. This is in radical contrast to states of mania, the opposing “up” state in bipolar disorder to depression’s “down”, where people report a radically accelerated temporal experience [81]. 

Phenomenological studies also connect these changes in the rate of temporal flow in the present experience to changes in how depressed people relate to the past and future. The guilt-ridden past exerts a dragging effect on an inexorable, unchangeable, and meaningless present experience because the future is “blocked” by feelings of hopelessness [82]. However, these phenomenological distortions of slowing in the present (i.e., dilation), cycling over the past (i.e., rumination), and blocking the future (i.e., hopelessness) do not seem to affect “intersubjective” time perception abilities [9,82]. As with psychomotor retardation, time dilation in depression is distinct from time perception as typically measured in laboratory experiments [83]. Instead, it is more related to the phenomenology of the narrative self over the extended time periods of the past, present, and future. In fact, time perception and passage over time judgements are only aligned over extended durations as shown in ecological experiments using experience sampling methods [84]. The relationship between time perception and the passage of time in depression is itself temporal in nature, specifying the long and slow nature of depressive time dilation.

To summarise, depressive time dilation is distinct from either psychomotor retardation (physiological slowing), or a slowed internal clock used for judging the intervals of clock time (cognitive slowing). The mechanistic explanation of slowing under MG provided above in terms of power calculations in a physical gravitational field therefore do not apply. Consequently, the more-intuitive physics of Newtonian [12] gravity no longer applies because the strength of the depressed MG field requires a more clinically precise explanation. The non-intuitive physics of Einsteinian gravity [76,77,85] is needed instead because general relativity deals explicitly with the phenomenon of *gravitational time dilation* (explained in detail below) that is absent from Newton’s formulation. Space and time are two aspects of the same thing in general relativity (i.e., spacetime) and gravity is caused, not by a force exerted between two masses as above, but rather by *spacetime curvature.* The next sections will focus on MG in depression as an analogue of spacetime curvature, both in the felt sense of distorted space and time, and also the curvature of underlying spatiotemporal neural dynamics.

## 6. Mental Gravity in Depression as Spacetime Curvature

### 6.1. Spacetime Curvature in General Relativity

According to general relativity, mass and energy curve spacetime around them so that the path of moving objects, including massless particles of light, bend in the presence of a gravitational field [86]. In our everyday experience, we observe this bending as an acceleration towards the surface of the Earth. Newton [12] theorised that a gravitational force was actively pulling objects down, but that is not the case according to the more precise general relativity. Acceleration is just the result of objects falling in a straight line through curved spacetime. According to Einstein [75], a gravitational field is a field of acceleration equivalent to spacetime curvature. Just as the vestibular organs cannot distinguish between gravity and linear acceleration, our everyday feeling of being “weighed down” on the Earth’s surface is difficult to separate from the feeling of being “accelerated up” away from the surface (cf., the equivalence principle between inertial and gravitational mass) [75]. The experiments of grip force in varying conditions during parabolic flights show that people can differentiate between acceleration and gravity [87,88], but this does not completely discount the fact that gravity feels like an acceleration.

In addition to objects following accelerated but straight paths through curved space (called geodesics), time also flows at different rates depending on an object’s location within a gravitational field. From the point of view of an external observer, time always flows more slowly for objects (i.e., clocks) that are closer to a massive object compared to objects further away. Time also flows more slowly when the massive object is larger and the gravitational field is stronger. This phenomenon is called gravitational time dilation and it has been experimentally proven that a clock in the near-Earth orbit ticks more quickly than a clock on the surface of the Earth, a critical consideration when calibrating orbiting satellites and planning astronautical missions [86]. Whereas Newton [12] proposed a “clockwork” universe where all events unfold according to absolute time, Einstein [75] showed that time and space are not absolute but *relative* to the local context of mass (and energy via the mass–energy equivalence, *E* = *mc*^2^). 

General relativity defines a gravitational field according to the reciprocal relationship between mass–energy and spacetime, expressed mathematically as:(5)$$g_{\mu \nu }=T_{\mu \nu }$$

Here, spacetime curvature is represented by the left-hand term, called the Einstein tensor, and the matter–energy contents are represented by the right-hand term, called the stress–energy–momentum tensor. This formula shows that both are mutually dependent—matter and energy tells spacetime how to curve, and spacetime tells matter and energy how to move [86]. In the MG theory of depression, the argument is that the high concentrations of autobiographical information (mental mass) and self-related attention (mental energy) is accompanied by distortions in the cognitive representation and subsequent experience of the self in space and time. Part of the MG effect is explicable in terms of the gravity’s effects on the body, as above, but the explanation also goes deeper to consider the underlying general relativity notion of spacetime curvature.

### 6.2. Analogy between Mental and Physical Time Dilation

By extending the behavioural analogy between physical and social–emotional environments in MG simulation, the common experience of time dilation in depression [9] is proposed here as a mental analogue of gravitational time dilation in general relativity. A visual depiction of the MG field is needed to illustrate the mechanics of this explanatory model. Figure 1 shows the self-axis extending from a central DMN point of narrative gravity to the self-world boundary which is mediated by the extension of the salience network’s present experience [65]. The side view compresses one of the three spatial dimensions to depict the strength of the gravitational field as a literal depression in the 2D fabric of space (plus one dimension of time). The mass causes a gravitational gradient of descent towards its centre, much like a massive ball will depress a 2D rubber sheet.

This heuristic depiction illustrates how the increased mass at the brain’s DMN centre of narrative gravity causes greater spacetime curvature in the MG field and increases the tendency of the salience network’s attentional resources to be directed inwardly. Time dilation is common in depression because the MG field is the strongest. Grave situations may also involve time slowing down, such as during traumatic accidents [89], but these experiences of acute panic or anxiety are typically transient compared to depression, and so the MG field is not as strong.

According to Figure 1, the MG field is so strong in depression and spacetime and so curved that the self is all but attentionally secluded from the outside world. Spacetime curvature may be temporarily strong in anxious states, but the self is still actively connected to the world by virtue of theirs being a “grave” situation to respond to. The self may be under threat during the “falling” MG state, but the depressed self is a threat to itself in the self-reinforcing “fallen” state.

In this schematic, depressive time dilation occurs because the salience network is directing attention “down” into the DMN’s strong MG field where the gravitational time dilation effects are sufficiently strong to be *observable*. Attending to self-related information causes a subjective slowing in the passage of time from the point of view of the salience network (i.e., the experiential self in the present moment), just as attending to a physical clock in a strong gravitational field will seem to tick more slowly than a clock in a weak gravitational field. The salience network mediates MG time dilation through arousal and attentional effects [90], and the DMN mediates the strength of the MG field by comprising the mental mass at the brain’s centre of narrative gravity.

### 6.3. Spatial Distortions in Depression

Depression is associated with distortions of space as well as time. Space is also distorted in depression at a very basic perceptual level. Depth perception tends to be diminished in depression due to reduced stereopsis [91]. Depressed individuals tend to report a “flatter” perception of objects, themselves, and other people [92]. Beyond perception, depressed individuals also use spatial metaphors to characterise their experience as “ENCLOSED SPACE”, “BOUNDED SPACE”, or “DARK CONFINED SPACE” [36], which imply a sense of spatial curvature as closedness or boundedness. Importantly, the DMN is implicated in spatial navigation as well as other functions associated with MG including autobiographical memory, prospection (future projection), and self-referential goal-directed tasks [93,94]. 

Phenomenological studies also suggest that depressed individuals tend to feel that external objects are out of reach, dull, or distant as though the self-world boundary has expanded [95]. Compared to anxiety, depression symptoms are associated with a reduced “anticipatory auditory looming bias”, meaning that depressed individuals tend to not perceive objects approaching as quickly or as closely, which reflects both a physical and cognitive disengagement from their surroundings in the physical and social space [96]. Depression distorts phenomenological space by creating a sense of separation, seclusion, and of being “locked in” from the external environment [97,98].

### 6.4. Combined Spacetime Distortions in Depression

A central feature of general relativity is the fact that space and time are combined into one four-dimensional field called spacetime which “curves” in the presence of mass or energy. For the analogy between general relativity and MG in depression to be valid, then it is not sufficient to show that space and time are distorted independently, as per the two preceding sections. Instead, one must show how subjective space and time are merged mentally and thus affected simultaneously. The concept of psychological distance merges mental space and time. Times, places, people, and counterfactual alternatives are mentally construed to exist at different distances from an egocentric reference point of one’s experience in the “here and now” [99]. Time, space, social distance, and hypotheticality each represent different “dimensions” of psychological distance [100]. Temporal distance is thus a form of a merged mental spacetime that construes the past, present, and future as different points along the time dimension. 

Temporal distance studies show that depression makes the past seem closer to, and the future more distant from the present [101]. Depressed people therefore feel spatially more distant from the external world [95] while also feeling that the future is more distant, showing a strong convergence between space and time in terms of physical and psychological distance. Spacetime curvature also manifests in non-linear memory recollections. A qualitative interview study by Habermas et al. [102] found that the tendency to feel “stuck” in the past in depression is accompanied by a tendency to order past autobiographical events non-linearly, grouping events thematically instead of temporally. The mental timeline of events in autobiographical episodic memory is not structured according to the linear ordering of memory encoding in physical time, but rather by the order of events in terms of psychological distance. From the egocentric point of view, objects that are thematically related are represented as being psychologically closer, and so events are recalled according to the subjective, psychological spacetime order instead of the intersubjective, physical time order.

### 6.5. Curvature of Mental Spacetime

Cognitive models also merge time and space via common computational processes. Van Wassenhove [103] proposed an amodal representational space common to the magnitudes of space, time, and numerosity, which is supported by psychophysical studies demonstrating the common processing of time, number, and length [104]. The principle of MG posits that mental activity (e.g., computation, cognition, representation, or information processing) occurs in a mental spacetime that curves in the presence of mental mass or energy. Stocker [105] theorised about such a “cognitive spacetime” but thought the concept of curvature was irrelevant. The principle of MG instead proposes that cognitive spacetime is not a flat, featureless, static background that serves the cognitive function of a “Cartesian theatre” where events simply unfold in *real* time [106]. Instead, mental spacetime is an active constituent of, and contributor to, cognitive and conscious activity precisely because it is not a fixed structure defining absolute space and time. Like physical spacetime, cognitive spacetime is governed by the principles of relativity, and so space and time are relative to the mass–energy context and vice versa. This reciprocal co-dependence is expressed mathematically in Equation (5) as the left spacetime term *equal to* the right energy–mass term. Cognitive spacetime therefore responds flexibly to mental matter–energy *while also actively shaping* those mental contents. Importantly, the curvature of mental spacetime can also be instantiated in the brain via the curvature of spatiotemporal neural dynamics.

## 7. Curvature in the Brain’s Spacetime

Whereas Newtonian mechanics was used above to model the intuitive physics of MG simulation (i.e., feeling down, low, heavy, and slow), the more complex mathematics of Einstein’s [95] general relativity can be used to model how non-intuitive gravity-like experiences emerge in depression. Le Bihan [10] proposed a relativistic model of neural activity which, like Einstein’s [95] theory of physical gravity, focuses on the central notion of *curvature* of the brain’s “spacetime” or “spacetime connectome” [107]. The framework, called relativistic pseudo-diffusion, proposes that the brain’s four-dimensional structure (three space dimensions and one time) is prone to a “functional curvature generated by brain activity, in a similar way gravitational masses give our 4-dimensional Universe spacetime its curvature” (p. 1) [10]. 

Le Bihan’s [10] model depicts neural activity propagating through a brain network in the same way that matter or energy propagates through physical spacetime. Neural activity diffuses along trajectories called “brainlines”, which are shaped via the structure and activity of the brain’s spacetime. Diffusion occurs at different rates depending upon the amount of neural activity (i.e., energy) and grey-matter density (i.e., mass) at specific network nodes, as well as the structure of white-matter interconnections (i.e., spacetime field). A greater density of nodes and higher node activity lowers the rate of diffusion of neural activity in the same way that increased mass–energy dilates time according to general relativity.

Of course, time and space are intrinsically linked in general relativity and so this dilation of time is also a result of the spatial distribution of the spacetime connectome’s network nodes and neural activity. In relativistic pseudo-diffusion, neural activity propagates out from the source through a spatially distributed “map” of the network nodes, traversing the topographical “peaks” and “valleys” of lower and higher density/activity, respectively. As per the side view of Figure 1, the connectome’s valleys are areas of slower propagation analogous to gravitational time dilation. Since gravity is a field of acceleration according to Einstein’s [75] general relativity, neural activity decelerates in the lower “valley” areas of brain spacetime that have higher energy or density (mass). In some sense, the propagation of neural activity *gravitates* towards depressions or valleys in the brain’s spacetime field. Le Bihan [10] uses Einstein’s field equations to model this process mathematically, as shown below:(6)$${R_{\mu \nu }-\frac{1}{2}Rg}_{\mu \nu }={-\kappa T}_{\mu \nu }$$

Similar to Equation (5), the left-hand terms represent spacetime curvature via the Einstein tensor and the geometric curvature of spacetime (*R_μv_* denotes the Ricci tensor, *R* the scalar curvature [0 in the absence of node activity in “flat” Minkowski spacetime], and *g_μv_* the brain spacetime Einstein metric tensor). The right-hand terms describe neural activity and density in a region of spacetime responsible for the curvature (κ a constant and *T_μv_* the stress–energy–momentum tensor [*T*_00_ represents local node activity and other terms denote activity in the spacetime surrounding the nodes]). Solving this field equation entails finding a metric tensor *g_μv_* of the brain’s spacetime curvature corresponding to the configuration of neural activity/density as defined by *T_μv_*. Equation (6) can thus be simplified to Equation (5), as shown below:$$g_{\mu \nu }=T_{\mu \nu }$$
which conveys the essential property of Einstein’s [75] theory of gravity—namely that spacetime (*g_μv_*) tells matter and energy (*T_μv_*) how to move, while matter and energy tells spacetime how to curve [86]. Gravitational time dilation is time flowing relatively slowly within the brain’s lower spacetime “valleys” of higher node density and activity and relatively quickly in the “higher peaks” of lower density and activity.

### 7.1. Time Dilation in Le Bihan’s Relativistic Pseudo-Diffusion Framework

Although not addressed by Le Bihan [10] specifically, time dilation in depression can be explained according to the relativistic pseudo-diffusion framework very straightforwardly. Consider that the DMN mediates self- or internally oriented attention while the central executive network mediates externally oriented attention [108]. Neuroimaging studies of depression suggest that, in the resting state, the DMN shows increased intrinsic activity and functional connectivity, whereas the central executive network shows decreased activity and connectivity [109]. According to Le Bihan’s [10] relativistic framework, neural activity in depression will diffuse more slowly through the DMN than the central executive network, meaning that internal or self-related attention and information will be flowing relatively slowly compared to external or world-related attention and information. This imbalance is the primary cause of time dilation in depression which, according to the relativistic pseudo-diffusion framework, is precisely analogous to gravitational time dilation.

Low central executive network activity and connectivity compared to the DMN can distort space as well as time through relativistic pseudo-diffusion. The curvature of mental spacetime can occur because the nodes of the central executive network mediate temporal, spatial, and social dimensions of psychological distance [110]. Whereas medial prefrontal nodes of the DMN mediate mind-wandering, the lateral prefrontal nodes of the central executive network mediate cognitive control [111]. Reduced central executive network connectivity or activity resulting in a loss of control constitutes the spatial separation, seclusion, or sense of being “locked in” in depression [97,98]. This central executive network is also highly integrated with the DMN and plausibly implicated in the breakdown of executive control over self-related thoughts and actions in depression that manifests as “sticky” autobiographical memory and prospection [112]. 

Importantly, the intuitive physics of falling MG is also consistent with a felt loss of control [30] and suppressed central executive network activity of depression [113]. The depressed individual feels less able to influence the world through planned, coordinated, and controlled behaviour and so their global mood (i.e., long-term average of the local emotional states) expresses this feeling using the template of falling that constitutes MG. Other intuitive gravity-like behaviours follow from this core belief of depression as falling to a descended state of being, giving the impression or feeling of being heavy, low, down, and slow. At a deeper computational, phenomenological, and neurobiological level, however, all of these surface-level impressions stem from an underlying problem with the way a person experiences time and space, the past and the future, and themselves and the world. The relativistic pseudo-diffusion framework offers a plausible neurobiological substrate for MG processes and experiences couched in the mathematics of general relativity [10]. Understanding how the brain’s spacetime manifests in depression could facilitate the earlier detection and perhaps prevention of depression, especially in high-risk groups such as children exposed to adverse experiences or trauma. Risk factors, such as changes in brain development, could show up as changes in the brain’s spacetime topography. These changes could serve as an early biomarker of depression risks that is more specific and sensitive than the current knowledge allows with respect to reduced cortical and subcortical grey-matter volume [114,115,116,117,118] or structural connectomics [119].

### 7.2. The “Common Currency” of the Brain’s Spacetime

The relativistic pseudo-diffusion framework can be used to model more than just depression. Le Bihan [10] uses it to model the temporal disordering of events in schizophrenia, resulting in confusions about internal versus external sources of information and hallucinatory symptoms. Combined spatial and temporal distortion as the basis for mental disorder mirrors Northoff’s [109] “spatiotemporal psychopathology” approach to depression. According to this model, depression arises because the DMN resting state activity and global functional connectivity are both abnormally high and new information is integrated more slowly:

In the healthy subject, both world- and self-related time usually correspond to a high degree and thus synchronized. This though changes in MDD where self-related time becomes slowed or decelerated compared and henceforth relative to world-related time […] Increased functional connectivity across the whole brain means that neural activity in specific regions (as triggered by either internal or external stimuli) can no longer change as much […] The abnormal increase in temporal continuity of the brain’s overall resting state activity is especially fostered via the infraslow frequency fluctuations showing increased functional connectivity (as measured in fMRI) because their extremely long phase durations (up to 100s) contribute strongly to an abnormally high degree of temporal continuity. (pp. 7–8).

Northoff’s (2016) passage covers key elements of the MG theory of depression: (1) the distorted sense of self as desynchronised from the world; (2) the slowing of self-related time; (3) the altered network function and connectivity in the DMN; and (4) an enhanced continuity or inertia due to a lower rate of change. According to Northoff [120], space and time are the “common currency” of all neural, cognitive, and phenomenological manifestations of depression in terms of the curvature of underlying neural spatiotemporal dynamics centred on the DMN, the central executive network, and the salience network [121,122,123]. Importantly, aberrant salience network–DMN/central executive network connectivity has been linked to depression severity [124] and all three regions mediate risk and resilience to the depressive disorder [125], meaning that Le Bihan’s [10] general relativity model completes the “common currency” of spacetime across phenomenological, cognitive, and neurobiological domains of psychological explanation. The phenomenal consciousness of those diagnosed with depression is fundamentally altered with respect to the experience of the self in space and time, meaning they also think differently about space and time in ways that map onto the altered spatiotemporal organisation of their brain. 

## 8. Spacetime Curvature in Predictive Processing

This MG principle is not alone in using curvature to model core aspects of mental processing. The underlying spacetime curvature of MG simulation is also explicable in predictive processing terms. From the Bayesian perspective, Friston et al. [126] referred to the “curvature of surprisal” and “surprisal gradients”. Recall that, throughout infancy and early childhood, the vestibular system and other graviceptive centres of the brain functionally integrate to distinguish the force of gravity from accelerations due to movement [15,127]. This internal gravity model contributes vital information to an intuitive physics of how and why objects fall and feel heavy [78], and MG simulation simply decouples strong, top–down expectations or gravity priors [17] from bottom–up sensory feedback [16].

If depression is both an increased spacetime curvature and an increased weighting of top–down gravity priors, then the principle of MG can be translated into the language of the “curvature of surprisal” used by Friston et al. [126]. They argue that self-information creates negative curvature (i.e., a depression, downwards and inwards) in the information geometry or state space that shapes the dynamics (i.e., trajectories) of perception and action. These curved trajectories are analogous to the geodesics that matter and energy travel along in a gravitational field. The MG theory of depression also stipulates attention is directed inwards and downwards (i.e., negative curvature) in depressed mental states with one key distinction. Whereas transient emotional states refer to the precision of prior beliefs about the consequences of perception and action, mood is the long-term average of the emotional states, with depression representing suboptimal long-term fluctuation [128]. Mood is therefore related to the so-called “hyperpriors” acting at a deeper-level of the predictive hierarchy over longer timescales than emotion [129]. 

Curvature on account of mood is therefore not the same “curvature of surprisal” that applies to a single trajectory of the present experiential self and the short-term emotional states. The long-term curvature of mood-states from hyperpriors is more akin to “hypercurvature” of *all possible* trajectories, meaning the curvature of the general spacetime/gravitational field as opposed to a single path through it. Another way to say this is that the brain’s DMN centre of narrative gravity (the self) is not just one of the multiple *local minima* in the curved manifold of the probability state space (time), but rather a *global minimum*. 

Sengupta et al. [130] expanded on the notion of curvature in probability state space to argue that “attention is a force that manifests from the curvature of information geometry, in exactly the same way that gravity is manifest when the space–time continuum is curved by massive bodies” (p. 7). According to the principle of MG, attention is not the only “force” that can create curvature. Mental mass also curves subjective spacetime in deeper, more global levels of the predictive hierarchy over the longer-term averages of experience and behaviour (e.g., long-term autobiographical memory, mood, cognitive development, and the diachronic self). 

Sengupta et al. [130] also connected the notion of spacetime curvature of information geometry to the underlying neural dynamics via so-called “neuronal gauge theories”, which link the activity of neural dynamics over a vast range of evolutionary, developmental, and perceptual timescales. Gauge theories are physical theories that describe the interactions between spacetime fields and matter or energy, defining properties that are invariant under the changing reference frames and under the conditions of curvature. Einstein’s [75] general relativity is one example of a gauge theory in physics, and the vestibular system also perceives gravity by transforming sensory information between body-centric and other non-body-centric reference frames [40,127,131]. Given that the reference frame transformation of spatiotemporal information is common to gauge theories, general relativity, and the perception of gravity, there may be a straightforward way to apply the principle of MG as a neuronal gauge theory in relation to the perception of gravity and various conscious states, including mood as a prime example.

The principle of MG could serve as a neuronal gauge theory linking the moment-to-moment experience of depression (and other manifestations of MG) to human lifespan development and cultural or biological evolution [132,133]. The MG theory of depression covers multiple timescales of gravity-like experiences from the development of “outside-in” empirical priors to the long-term signs and symptoms of depression, listed in Table 1, that manifest as a feeling of personal descent [36], perceived bodily heaviness [23], subjective time dilation [9,80], an altered sense of self in relation to narrative gravity [37,65], and distorted neural spatiotemporal dynamics [109,134,135]. More work is needed to translate the principle into a more rigorous conceptual framework, such as Bayesian Mechanics [136], via an implementation framework such as relativistic pseudo-diffusion which uses the mathematics of general relativity to model the mental energy–matter contents and mental spacetime [10].

## 9. Summary and Implications for the TTC

The MG principle contends that the mind uses physical gravity as a mental model, isomorphism, or simulacrum to express the relation between the inner self and the outer world in terms of “UP”-ness and “DOWN”-ness (i.e., UVV). The simulation of increased gravity characterises a continuum of MG states which includes depression as the paradigmatic example of being down, low, heavy, and slow. Simulated graviception is mediated via the anterior insula cortex of the salience network but revolves around the centre of MG identified with the DMN as the mental mass defining the strength of MG for an individual, which is equivalent to DMN activity as the brain’s narrative centre-of-gravity. The mechanics of MG in depression is modelled using the basic Newtonian physics of gravitation to show how the perception and (hyper)embodiment and increased bodily mass can cause other feelings associated with being down, low, heavy, and slow. The general relativity physics of gravity can also be used to model spacetime curvature in depression, particularly gravitational time dilation as a property of MG analogous to subjective time dilation (i.e., the slowing of temporal flow in conscious experience), but also the curvature of the brain’s spacetime and cognitive spacetime.

The principle of MG has profound implications for the TTC. Northoff and Huang [1] suggest that “temporo-spatial alignment is altered in psychiatric patients (e.g., schi- zophrenia, depression and bipolar disorder) corresponding to abnormal form of consciousness” (p. 642). As opposed to the short-term alignment for specific local contents of consciousness, the authors contend that a long-term temporo-spatial alignment establishes a “world-brain relation” that is centred on embodiment and the socialisation of conscious states. This is in accordance with Whiteley’s [6] view that a depressed mood is a distinct global state of consciousness that involves an experiential “shift” to a world that is seemingly detached from reality and other people. 

Long-term temporo-spatial misalignment in depression is associated with the observation that DMN activity occurs over the longest timescale in the infraslow frequency range [137]. Increased DMN activity or connectivity in depression serves as the TTC neural correlate of mental mass and a neural mechanism for determining the strength of the MG field. The TTC can connect the idea of a narrative centre of gravity to the activity of DMN and the cortical midline structures associated with long-term temporo-spatial alignment and, thus, the notion of the curvature of spacetime in conscious experience. In concert with other theories of cognitive [105] and neurobiological spacetime [10], the TTC can also work towards the “common currency” approach to space and time that also potentially connects the TTC to predictive processing frameworks such as free energy [138,139], neuronal gauge theories [130], and active inference accounts of depression [140].

In sum, the principle of MG provides the TTC with a way to incorporate the structure of the world into the structure of the brain, conscious experience, and thought. It gives the up/down dimension of space, as defined by the gravitational field, a unique status that is connected to both our embodied interaction with the physical world, and also the inverse, reflective, emotional but still embodied experience of ourselves.

## Figures and Tables

**Figure 1 entropy-25-01275-f001:**
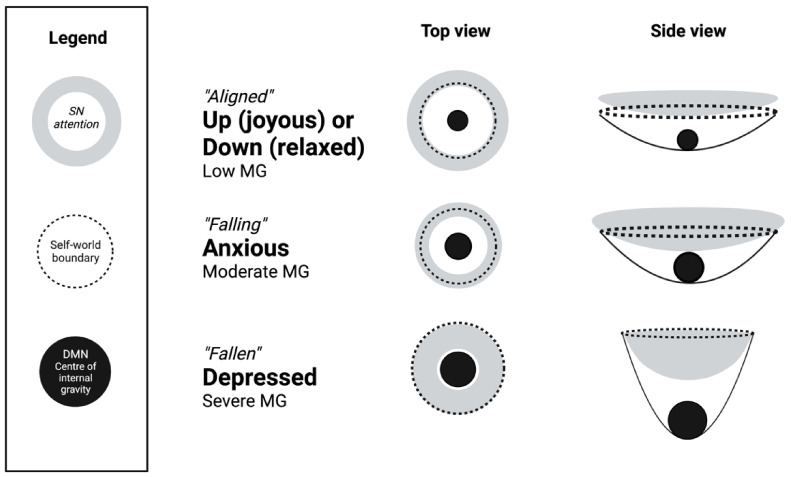
Schematic of MG field states in terms of the self-axis extending from DMN centre to salience network limit at the self-world boundary. The mass at the DMN centre determines degree of spacetime curvature and the tendency to draw salience network attentional resources inwards and down in anxiety and depression, i.e., the strength of the MG field.

**Table 1 entropy-25-01275-t001:** Similarities between the aspects of physical gravity (left) and mental gravity in depression (right).

Physical Gravity	Mental Gravity in Depression
Newtonian Gravity and Intuitive Physics	Simulation of Intuitive Physics
*Down or low:* objects fall down towards the centre of gravity but require energy to ascend or escape gravity’s “pull”. Equivalent to height variable (*H*) in potential energy and power calculations.	*Feeling “down” or “low”:* the feeling of personal descent and of being stuck in a low state of being (i.e., effortless descent but effortful ascent).
*Heavy:* the same physical mass weighs more in a stronger gravitational field. Equivalent to mass variable (*m*) in gravity, energy, and power calculations.	*Feeling “heavy”:* the feeling of increased bodily weight.
*Higher energy costs and restricted motion:* due to increased weight in stronger gravitational fields, it either takes more energy to move or else movement takes longer.	*Feeling fatigued and “slow”:* psychomotor retardation as the result of increased mental effort/energy required to move the body, causing fatigue (or compensatory agitation) and also a sense that movement is restricted.
**General relativity**	**Experience of non-intuitive physics**
*Spacetime curvature:* acceleration in a gravitational field occurs when objects move along straight paths through spacetime curved by mass–energy causing gravitational time dilation, length contraction, and curved geodesics (trajectories).	*Time dilation:* passage of time in conscious experience is slow and expanded (i.e., dilated). *Spatial contraction:* the self and world are experienced as flat. *Spacetime combined:* psychological temporal distance is altered, cognitive spacetime is distorted, events are disordered, and trajectories of neural spatiotemporal dynamics are curved.

## Data Availability

Not applicable.

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
