# Peer review of "Mental Gravity: Depression as Spacetime Curvature of the Self, Mind, and Brain"

_entropy, 2023, doi:10.3390/e25091275_

Round 1
Reviewer 1 Report
I should say from the outset that, although I am very familiar with how the brain processes gravity from neurophysiological and behavioral standpoints in order to perform actions, the material presented in this ms seemed extremely abstract and dissociated from reality.
I read the ms a second time. I then realized that it actually shares a lot of concepts with my research. This was fascinating.
Although a bit outside the scope of this assessment and from a more philosophical point of view, we know that exchanges between different disciplines is often difficult because of different language codes and approaches. However, when aa dialogue and mutual comprehension is possible, treasures can be found, new ideas can emerge, and new collaborations can be fostered.
Please find below some comments and questions left to the author’s appreciation.
Page 4. Typo. “The theory of MG is connects the experience...”.
Page 5. First paragraph under “Overview of the Theory of Mental Gravity”.
In the neuroscience field, the way human beings (and primates) move in their environment is very well explained and described through the concept of internal models. Briefly, we have a target or an objective in mind. This is the planning step. Then, from that objective, the brain finds the optimal commands required to reach that goal. This is the role of the control policy and take into account some cost function (energy, time, etc). Once the motor commands are calculated, they can be issued in the environment where the action takes place. In parallel, a copy of these motor commands (efference copies) are route inside the brain into a forward model that acts like a simulator of the body in its environment. The forward model calculates the (sensory) consequences of the action. This is a prediction. The brain can then compare these predictions with the real sensory consequences measured by (delayed) biological feedback. If both match, this means the actor has a good internal representation of the dynamics of his/her body, the environment and their interaction. If there is a mismatch (error), then some correction must take place somewhere. The task of the brain is to infer where the error comes from (perception? Cost function? Control policy? Forward model?). I think that MG is a subset of this concept as it focuses only on one, yet critical, aspect that is gravity. The author should broaden a little the scope of this opening and consider other key references that paved the route toward the theories of internal models (Angelaki et al., 2004; Bursztyn et al., 2006; Kawato, 1999; Mcintyre et al., 2001; Merfeld et al., 1999; Wolpert et al., 1998).
Page 6. The author mentions that depressed patients feel “down” or “slow”. When movements are performed in hypogravity (weightlessness at the extreme), they are also slowed down (Mechtcheriakov et al., 2002; White et al., 2008).
The next paragraph (“Continuum of MG Experiences”) addresses the continuum between these two extremes (e.g. UP vs. DOWN). However, an infinite number of intermediate states exist between these extremes. A hallmark of the optimization of the gravitational field by the brain is the presence of an asymmetry in peak velocities in reaching movements. In horizontal movements, the peak velocity occurs exactly in the middle of the trajectory. However, in upward vertical movement, this peak occurs earlier than 50% (and later in vertical downward movements) (Papaxanthis et al., 1998). There is a continuum of asymmetry between horizontal and vertical movements (Gaveau et al., 2016).
Page 7 (top). I don’t understand the following sentence (elevation?). “The theory of MG reflects our experience of physical gravity in that it is not just about elevation but alignment to the gravitational field.”
Page 7 (The Internal Gravity Model). I would stress the fact that gravity, unlike other senses, has no dedicated organs. Instead, graviception results from a distributed integration process including the vestibular system, vision, proprioception, somatosensory, and even visceral information. See recent review about this (White et al., 2020).
Page 8 (Mental Gravity as Simulated Graviception). An important part of the motor control/learning field of research in motor imagery. Mentally imagining performing an action partly activates neural circuitries involved in motor learning. An intriguing question is whether one is able to realistically simulate an action that he or she has never performed in the past. In particular, can the effects of weightlessness on the body and the environment be extrapolated mentally even if the person has never experienced that context (Rannaud Monany et al., 2022)?
Page 11. “template” is often referred to as calibration in the neurophysiology field.
Pages 11-12. Paragraph about mental mass. On the Earth, mass is perceived through its weight (and less to its inertia). The analogy is that depressed patients feel heavy. Does this imply that they could be cured in weightlessness (or bedrest, more pragmatically)?
Page 12. Typo. “evident in in”
Page 12. Comment. Note that gravity has not been demonstrated to be a force yet. Its effects act like a force. This fourth fundamental interaction still fails to be described coherently in quantum mechanics with the three others. No particle (hypothetical graviton) has been discovered (yet?).
Table 1.
- Please capitalize or not after “:” in the table (just be coherent).
- Source of gravity should be center of gravity
- (H) in potential energy
- Higher energy costs and slower motion, are not only due to increased mass and hence, increased weight. In rhythmic movements, the best way to pick up the optimal movement frequency or tempo is to choose the resonant frequency as it optimizes the energy exchanges between the environment and the system. A simple mathematical pendulum has a period that depends on the length of the string (the longer the string, the slower the period) and gravity (the smaller gravity, the longer the period), but NOT mass. Therefore, a slowing down reflects an optimization of energy because of a decrease of gravity (length constant), whatever the mass… (Boulanger et al., 2020, 2021; White et al., 2008)
- Feeling heavy is also reported after a difficult digestion…
- Time dilation: the passage of time is also altered (sped up) when we do something that needs concentration.
Page 15. Naïve question: g has no repulsive force. Is there any analogy between mental states and repulsive forces? Making parallels with the electromagnetic force (that is attractive and repulsive) may be more complete in that sense.
Page 15. Formula 2: this is potential energy.
Page 18. In the expression “time is an eternity”, time is paused, not slowed down.
Page 19. “Just as the vestibular organs cannot distinguish between gravity and linear acceleration, our everyday feeling of being “weighed down” on the Earth’s surface is equivalent to the feeling of being “accelerated up” away from the surface (cf., the equivalence principle between inertial and gravitational mass; Einstein, 1920).”
From a theoretical point of view, I completely agree with this. However, experiments in parabolic flights have shown that people actually make a difference between acceleration and gravity. When you move an object, it generates a load force LF that depends on mass, acceleration and gravity: LF=mg+ma. That force varies, and one needs to continually adjusts his or her grip force to conteract it. The question is: what happens if we experimentally generate exactly the same LF but using different combinations of acceleration and gravity? A previous experiment demonstrated that the brain can differentiate the source of the LF to adjust GF appropriately (White, 2015; White et al., 2005).
General comment: there are too many acronyms in the ms.
Figure 1 (page 23) depicts deformation of the topology with mass. There is also a singularity in the sense that a collapse generates a point dimensionless with an infinite mass. Does this mean that death is an extreme outcome to depression, assimilated to a black hole? Anecdotally, experiments in hypergravity (>1g) can often be generalized from 1g (e.g. 1g+1g = 2g). However, what happens in weightlessness (0g) is a totally different story (1g-1g is not 0g). In tat sense, 0g is also a singularity.
Page 24. “Phenomenological studies also suggest that depressed individuals tend to feel that external objects are out of reach, dull, or distant as though the self-world boundary has expanded (Sass & Pienkos, 2013).” This also implies that more energy is required to reach them.
Page 24. Typo. “…that space and time a distorted…”
Page 25. “…manifests in non-linear memory recollections…”. Perception of time also varies with age.
I found the analogy with GR and the tensors quite far-fetched (poetry). How can these concepts an analogies help cure depressed patients?
Page 30. “The “common currency” of the brain’s spacetime”. It is shown that a healthy system (living or not) exhibits some inherent complexity. This is quantified by a fractal dimension. For instance, heart rate, although regular, should include some small variabilities to be healthy. In case of stress (fear or exercice), heart rate tends to regularize, which is not “good”. Is there also such complexity measurement in anxiety disorders? (Iglesias-Parro et al., 2016)
Angelaki, D. E., Shaikh, A. G., Green, A. M., & Dickman, J. D. (2004). Neurons compute internal models of the physical laws of motion. Nature, 430(July), 560–564. https://doi.org/10.1038/nature02754
Boulanger, N., Buisseret, F., Dehouck, V., Dierick, F., & White, O. (2020). Adiabatic invariants drive rhythmic human motion in variable gravity. Physical Review E, 102(6). https://doi.org/10.1103/PhysRevE.102.062403
Boulanger, N., Buisseret, F., Dehouck, V., Dierick, F., & White, O. (2021). Motor strategies and adiabatic invariants: The case of rhythmic motion in parabolic flights. Physical Review E, 104(2), 024403. https://doi.org/10.1103/PhysRevE.104.024403
Bursztyn, L. L. C. D., Ganesh, G., Imamizu, H., Kawato, M., & Flanagan, J. R. (2006). Neural correlates of internal-model loading. Current Biology : CB, 16(24), 2440–2445. https://doi.org/10.1016/j.cub.2006.10.051
Gaveau, J., Berret, B., Angelaki, D. E., & Papaxanthis, C. (2016). Direction-dependent arm kinematics reveal optimal integration of gravity cues. ELife, 5(NOVEMBER2016), e16394. https://doi.org/10.7554/eLife.16394
Iglesias-Parro, S., Soriano, M. F., & Ibáñez-Molina, A. J. (2016). Fractals in Affective and Anxiety Disorders. In A. Di Ieva (Ed.), The Fractal Geometry of the Brain (pp. 471–483). Springer. https://doi.org/10.1007/978-1-4939-3995-4_29
Kawato, M. (1999). Internal models for motor control and trajectory planning. Curr Opin Neurobiol, 9(6), 718–727. https://doi.org/10.1016/S0959-4388(99)00028-8
Mcintyre, J., Zago, M., Berthoz, A., & Lacquaniti, F. (2001). Does the brain model Newton ’ s laws ? 4(7), 693–694.
Mechtcheriakov, S., Berger, M., Molokanova, E., Holzmueller, G., Wirtenberger, W., Lechner-Steinleitner, S., De Col, C., Kozlovskaya, I., & Gerstenbrand, F. (2002). Slowing of human arm movements during weightlessness: The role of vision. European Journal of Applied Physiology, 87(6), 576–583. https://doi.org/10.1007/s00421-002-0684-3
Merfeld, D. M., Zupan, L., & Peterka, R. J. (1999). Humans use internal models to estimate gravity and linear acceleration. Nature. https://doi.org/10.1038/19303
Papaxanthis, C., Pozzo, T., Popov, K. E., & McIntyre, J. (1998). Hand trajectories of vertical arm movements in one-G and zero-G environments. Evidence for a central representation of gravitational force. Experimental Brain Research. Experimentelle Hirnforschung. Expérimentation Cérébrale, 120(4), 496–502.
Rannaud Monany, D., Barbiero, M., Lebon, F., Babič, J., Blohm, G., Nozaki, D., & White, O. (2022). Motor imagery helps updating internal models during microgravity exposure. Journal of Neurophysiology. https://doi.org/10.1152/jn.00214.2021
White, O. (2015). The brain adjusts grip forces differently according to gravity and inertia: A parabolic flight experiment. Front. Integr. Neurosci., 9(February), 1–10. https://doi.org/10.3389/fnint.2015.00007
White, O., Bleyenheuft, Y., Ronsse, R., Smith, A. M. A. M., Thonnard, J.-L. J.-L. L., & Lefèvre, P. (2008). Altered gravity highlights central pattern generator mechanisms. Journal of Neurophysiology, 100(5), 2819–2824. https://doi.org/10.1152/jn.90436.2008
White, O., Gaveau, J., Bringoux, L., & Crevecoeur, F. (2020). The gravitational imprint on sensorimotor planning and control. In Journal of Neurophysiology. https://doi.org/10.1152/jn.00381.2019
White, O., McIntyre, J., Augurelle, A. S. A.-S. A.-S. S., Thonnard, J.-L. J.-L. L., & McIntyre, J. (2005). Do novel gravitational environments alter the grip-force/load-force coupling at the fingertips? Experimental Brain Research, 163(3), 324–334. https://doi.org/10.1007/s00221-004-2175-8
Wolpert, D. M., Miall, R. C., & Kawato, M. (1998). Internal models in the cerebellum. In Trends in Cognitive Sciences (Vol. 2, Issue 9, pp. 338–347). https://doi.org/10.1016/S1364-6613(98)01221-2
Reviewer 2 Report
This is a very good paper on a difficult subject, and I mostly fully agree with the argument that aligns with my own thinking on these matters. There is only one but quite a significant issue to be addressed and corrected in my opinion - space is not two-dimensional, and the same goes for lived space, and also for deformations of space in depression. Live space-time in depression is curved beyond the UP and DOWN aspects; e.g. there is the remoteness of things, there is the flatness of affect, there are limitations of movement, there is the "pulling-back" which is both temporal and spatial (precisely as the author argues himself!); so there is a three-dimensional depth of the lack thereof in addition to the vertical line; the notion of the constricted temporal horizon is also related to space and "forward" looking (it is a horizon!), etc.
The author seems to be aware of all this but nonetheless when it comes to the meta-level presents the issue in only vertical and binary terms (e.g. p. 7, in the conclusions, in the abstract) as UP vs DOWN (and related terms).
In Table 1 for example the author mentions the experience of "flatness" in depression, but not the length contraction; also p. 23 speaks of flatness, and p. 24 of distance. I fully agree that "Temporal distance is thus a form of merged mental spacetime that construes the past, present, and future as different points along the time dimension" (p. 24) - but if space-time is four-dimensional, the lived space-time should be consistently presented as four-dimensional and not two-dimensional (at least when presented as a counterpart of GR).
This key comment actually favors the argument of the paper because I think it gives more than it claims to give, except that it is not clear on the meta-level of the presentation.
It is also worth mentioning that the argument on the lived unity of space-time was made years ago by Erwin W. Straus, one of the key phenomenological psychiatrists working on depression. The author would appreciate the arguments made by Straus, e.g. in the famous essay "The Forms of Spatiality". I finally recommend having a look at a 1946 essay by Straus that concerns the very same issue the phenomenological part of the paper is about - it is titled "Psychotic Disorders of Space and Time" (it concerns space-time in depression) and was recently published in Frontiers in Psychiatry (Moskalewicz & Fuchs, Psychotic Disorders of Space and Time, 11 April 2023, https://www.frontiersin.org/articles/10.3389/fpsyt.2023.1150005/full and https://www.frontiersin.org/articles/10.3389/fpsyt.2023.1150005/full#supplementary-material) .
I am sure the author will appreciate the points made by Straus on the unity of lived space-time and affect, which align with his own but are more phenomenologically refined.
Other issues:
-slowed felt-sense of time in depression is not a cognitive sense of time (p. 17) - a very good point!
-I think the capitalization of some concepts and quotes should be avoided for the sake of professionalism
- why speak of the "Theory of Mental Gravity"; it's not really a theory, rather a hypothesis at best, but I would say a "theoretical conception" or something along this line
Round 2
Reviewer 1 Report
The reviewer is very satisfied with this revision. The author has carefully taken the time to consider all questions and comments. The ms has improved in terms of visibility. Very nice piece of literature!
Reviewer 2 Report
Good luck with your work, I would also be happy to cooperate in the future.